# Approximate distance correlation for selecting highly interrelated genes across datasets

Qunlun Shen[1,2], Shihua Zhang[1,2,3,4] *

**1** NCMIS, CEMS, RCSDS, Academy of Mathematics and Systems Science, Chinese Academy of Sciences, Beijing, China, **2** School of Mathematical Sciences, University of Chinese Academy of Sciences, Beijing, China, **3** Center for Excellence in Animal Evolution and Genetics, Chinese Academy of Sciences, Kunming, China, **4** Key Laboratory of Systems Biology, Hangzhou Institute for Advanced Study, University of Chinese Academy of Sciences, Chinese Academy of Sciences, Hangzhou, China

* zsh@amss.ac.cn

**Data Availability Statement:** All relevant data are within the manuscript and its Supporting information files.

**Funding:** This work has been supported by the National Key Research and Development Program

## Abstract

With the rapid accumulation of biological omics datasets, decoding the underlying relationships of cross-dataset genes becomes an important issue. Previous studies have attempted to identify differentially expressed genes across datasets. However, it is hard for them to detect interrelated ones. Moreover, existing correlation-based algorithms can only measure the relationship between genes within a single dataset or two multi-modal datasets from the same samples. It is still unclear how to quantify the strength of association of the same gene across two biological datasets with different samples. To this end, we propose Approximate Distance Correlation (ADC) to select interrelated genes with statistical significance across two different biological datasets. ADC first obtains the $k$ most correlated genes for each target gene as its approximate observations, and then calculates the distance correlation (DC) for the target gene across two datasets. ADC repeats this process for all genes and then performs the Benjamini-Hochberg adjustment to control the false discovery rate. We demonstrate the effectiveness of ADC with simulation data and four real applications to select highly interrelated genes across two datasets. These four applications including 21 cancer RNA-seq datasets of different tissues; six single-cell RNA-seq (scRNA-seq) datasets of mouse hematopoietic cells across six different cell types along the hematopoietic cell lineage; five scRNA-seq datasets of pancreatic islet cells across five different technologies; coupled single-cell ATAC-seq (scATAC-seq) and scRNA-seq data of peripheral blood mononuclear cells (PBMC). Extensive results demonstrate that ADC is a powerful tool to uncover interrelated genes with strong biological implications and is scalable to large-scale datasets. Moreover, the number of such genes can serve as a metric to measure the similarity between two datasets, which could characterize the relative difference of diverse cell types and technologies.

of China [2019YFA0709501]; the Strategic Priority Research Program of the Chinese Academy of Sciences (CAS) [XDPB17], the Key-Area Research and Development of Guangdong Province (2020B1111190001), the National Natural Science Foundation of China [61621003]; the National Ten Thousand Talent Program for Young Top-notch Talents, the CAS Frontier Science Research Key Project for Top Young Scientist [QYZDB-SSW-SYS008] and the Shanghai Municipal Science and Technology Major Project [2017SHZDZX01] to SZ. The funders had no role in study design, data collection and analysis, decision to publish, or preparation of the manuscript.

**Competing interests:** The authors have declared that no competing interests exist.

## Author summary

The number and size of biological datasets (e.g., single-cell RNA-seq datasets) are booming recently. How to mine the relationships of genes across datasets is becoming an important issue. Computational tools of identifying differentially expressed genes have been comprehensively studied, but the interrelated genes across datasets are always neglected. Detecting of highly interrelated genes across datasets is hindered because the samples of them are always different and they could have different numbers of samples. To solve this problem, we present a new algorithm that can identify interrelated genes across datasets based on distance correlation. Our proposed algorithm is very efficient and works well in different technologies, i.e., RNA-seq, single-cell RNA-seq and single-cell ATAC-seq. Also, we found that the number of such highly interrelated genes can serve as a metric to measure the similarity between two datasets, which could characterize the relative difference of diverse cell types and technologies.

This is a PLOS Computational Biology Methods paper.

## Introduction

High-throughput sequencing technologies (e.g., RNA-seq, scRNA-seq, scATAC-seq) provide an unprecedented opportunity to analyze biological process with large-scale data. For example, The Cancer Genome Atlas (TCGA) profiles numerous cancers with large amounts of omics data [1]; The Human Cell Atlas (HCA) profiles transcriptomics of thousands to millions of cells at single cell level. Recently, scATAC-seq has been greatly improved in terms of cell throughput and sequencing efficiency to view chromatin accessibility [2]. Integrative and comparative studies of such data is becoming a key tool to decipher the underlying relationships among genes [3–6].

Differential analysis plays a vital role in comparative studies, and many methods like limma [7] and edgeR [8] have been put forward to identify differentially expressed genes between two different datasets [9]. However, such methods fail to identify genes with similar expression patterns between two datasets with different samples. Meanwhile, the problem of measuring the correlation between two genes in a single dataset or two multi-modal datasets from the same samples has been well studied and can be conducted using Pearson correlation coefficient, Spearman correlation coefficient, Kendall correlation coefficient and so on. However, these methods only capture linear dependence. Complex nonlinear dynamics exist widely in biological systems [10], which cannot be explained by simple linear relationship. More recently, the maximal information coefficient (MIC) [11] has been proposed to discover linear and non-linear dependency among the variable pairs in exploratory data mining, and detected a wide range of interesting associations between pairs of variables. For example, MIC has been applied to yeast gene expression profiles to identify genes whose transcript levels oscillate during the cell cycle [11]. It should be noted that the performance of MIC can be significantly reduced with a limited number of samples in practice [12].

However, these methods are incapable of quantifying the associations of genes across two different datasets from different samples, preventing us from finding genes with patially similar expression patterns under different conditions. Specifically, for a target gene, we would like

to measure the correlation of this gene across datasets, i.e., the correlation between two gene vectors with different samples (dimensions). This is a tough task since we don't have matched samples. Thus, the correlation is not supposed to rely on the order of the samples (order-free), which means this correlation is determined by the distribution of genes. The distribution here indicates that the expression of a gene across a given number of cells are considered as the expression observations of this gene in a given dataset. A few advances have been made to measure the correlation of two random vectors with diverse dimensions. For example, distance correlation (DC) [13] was designed based on the principle that the two random vectors are independent if and only if their joint characteristic functions are the same as the product of their marginal characteristic function. An unbiased sample estimation was constructed to calculate the distance correlation coefficient. Under the null hypothesis, there exists a statistic obeying $t$-distribution based on sample estimation of DC. So we can easily calculate the $p$ value of this hypothesis testing. This method views gene pairs with different dimensions as two gene vectors, and requires multiple observations to measure the associations between them, which makes it possible to compare gene vectors with different dimensions. However, we usually only have one observation or very few replicates for a target gene pair in real biological datasets. It is impossible to calculate DC directly under such circumstances. Therefore, quantifying the strength of association between genes across datasets remains an outstanding challenge.

To this end, we propose Approximate Distance Correlation (ADC) to robustly select genes with high interrelation across two datasets (Fig 1). ADC first obtains the $k$ most correlated genes for each target gene as its approximate observations, and then calculates the distance correlation (DC) for the target gene across two datasets. ADC repeats this process for all genes and then performs the Benjamini-Hochberg (BH) adjustment to control the false discovery rate (FDR). Extensive experiments with simulation data and four biological applications demonstrate that ADC can uncover the most interrelated genes, which illustrates strong biological relevance. Moreover, the number of similar genes selected could serve as an index to measure the degree of similarity across different cell types and technologies. Lastly, ADC can be applied to datasets ranging from thousands to millions of cells.

## Materials and methods

### Data and preprocessing

We applied ADC to four biological scenarios (Tables A-D in S1 Supplementary Materials). (i) We downloaded the gene expression data of 21 different cancers with more than 200 tumor samples for each on 20 March 2020 from http://gdac.broadinstitute.org. For each cancer type, housekeeping genes [14] and genes with no expression in more than half of the tumor samples were removed. After filtering, we log-transformed the expression with a pseudo-count 1 and perform $z$-score normalization for each gene.

(ii) We downloaded the scRNA-seq data of mouse hematopoietic cells from NCBI Gene Expression Omnibus with accession code GSE81682. Cells with less than 200 expressed genes and genes expressed in less than 3 cells were filtered. Also, cells with more than 25% mitochondrial genes present were filtered. We normalized the cells by size factor using the compute-SumFactors function in the scran package [15] and log-transformed the gene expression data with a pseudo-count 1. For each cell type, the top 1000 highly variable genes were selected with scanpy package [16] as the input to ADC. There are six cell types remained including hematopoietic stem cells (HSC), common myeloid progenitors (CMP), granulocyte-onocyte progenitors (GMP), megakaryocyte-erythroid progenitors (MEP), common lymphoid progenitor (MPP), lymphoid-primed multipotent progenitor (LMPP) with 323, 328, 123, 362, 368, 280 cells, respectively.

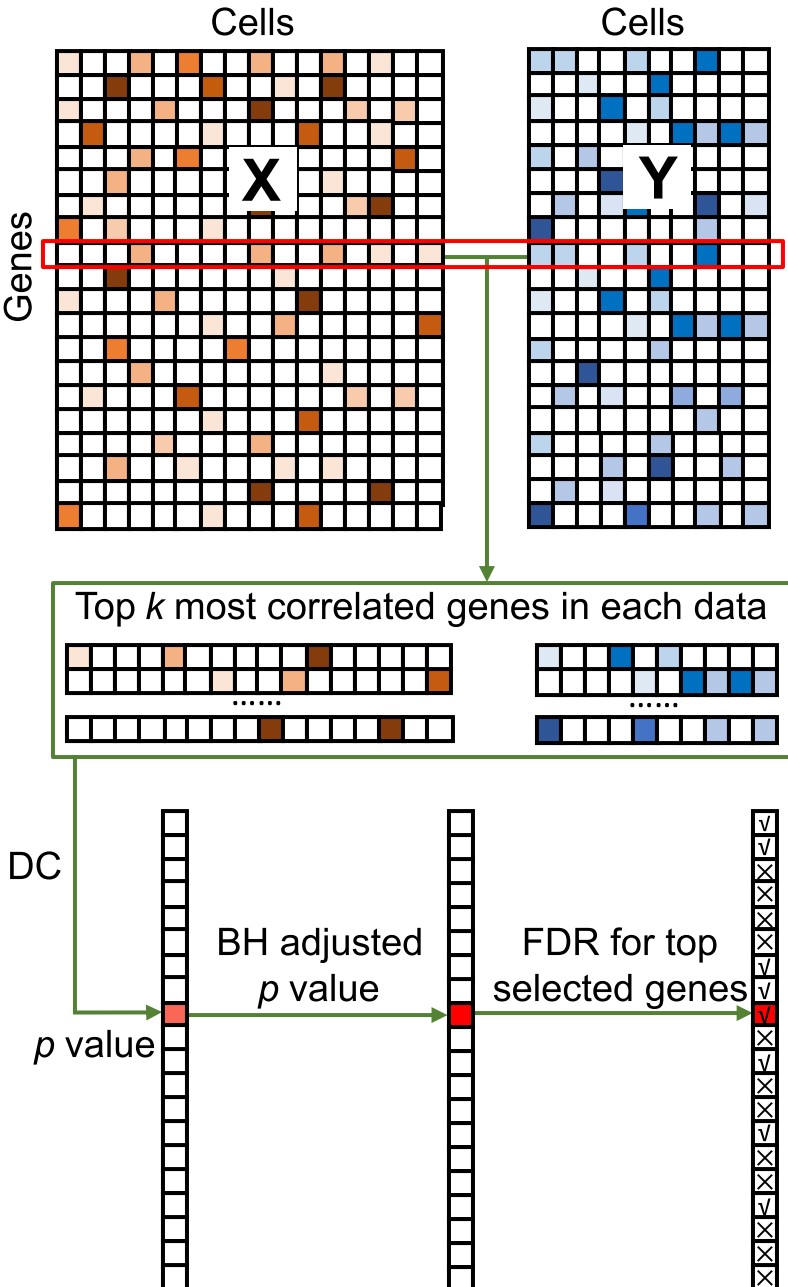

**Fig 1. Schematic diagram of ADC.** Single-cell gene expression data are used as an example to illustrate ADC. **X** and **Y** are data matrices with matched genes as the inputs of ADC. For each target gene, ADC selects *k* genes having the highest Pearson correlation coefficient with the target one to calculate the *p* value of DC. After that, ADC performs the BH adjustment to control the FDR and outputs the most highly interrelated genes.

(iii) We obtained five scRNA-seq datasets of pancreatic islet cells profiled by five different technologies including indrop, CEL-Seq2, 10X, CEL-Seq, Smart-seq2 respectively [17–21]. Housekeeping genes were removed [14]. We normalized the gene counts per cell with the scanpy package and log-transformed the expression with a pseudo-count 1 and the z-score normalization were also performed for each gene. Finally, these data consists of 8569, 4776, 2477, 1538, 3354 cells respectively.

(iv) We obtained the scRNA-seq data of human PBMC [22] and the scATAC-seq data [23]. The latter was transformed to gene activity matrix by the CreateGeneActivityMatrix function in the R package Seurat and the annotation file Homo_sapiens.

GRCh38.91. The preprocessing procedure was the same as (iii). Finally, these data consists of 8728, 2638 cells respectively.

## The ADC algorithm

Here, we have two biological data matrices $\mathbf{X} \in R^{p \times m}$ and $\mathbf{Y} \in R^{p \times n}$ with $p$ matched genes, and each row represents a gene while each column represents a sample (cell). ADC is designed to measure the interrelation for each gene between two datasets based on distance correlation (DC) and selects the most similarly interrelated ones (Fig 1 and Algorithms in S1 Supplementary Materials).

**DC.** DC is calculated based on distance covariance. Distance covariance is a method to measure the distance between the product of marginal characteristic functions of two random vectors $X \in R^m$ and $Y \in R^n$ and their joint characteristic function. It is defined as:

$$\mathcal{V}^2(X, Y) \quad = \|\phi_{X,Y}(t,s) - \phi_X(t)\phi_Y(s)\|_w^2$$
$$= \int_{\mathbb{R}^{m+n}} |\phi_{X,Y}(t,s) - \phi_X(t)\phi_Y(s)|^2 w(t,s) dt ds, \tag{1}$$

where

$$w(t,s) = \left(c_m c_n |t|_m^{1+m} |s|_n^{1+n}\right)^{-1}, \quad c_d = \frac{\pi^{\frac{1+d}{2}}}{\Gamma\left(\frac{1+d}{2}\right)}, \tag{2}$$

here $\Gamma(\cdot)$ is the gamma function and the weight function $w(t,s)$ ensures distance covariance is less than infinity. This definition is similar to that of the classical covariance and has a significant property, i.e., $X$ and $Y$ are independent if and only if $\mathcal{V}^2(X, Y) = 0$. Just as the standard definition of correlation coefficient, the DC is defined as:

$$\mathcal{R}^2(X, Y) = \frac{\mathcal{V}^2(X, Y)}{\sqrt{\mathcal{V}^2(X, X)\mathcal{V}^2(Y, Y)}}. \tag{3}$$

However, we don't know the exact distributions of $X$ and $Y$. If we have $k$ observations from them, we could use sample estimation to approximate DC. Let $\phi_X^k$, $\phi_Y^k$ and $\phi_{X,Y}^k$ denote the empirical characteristic functions of $X$, $Y$ and $(X, Y)$ (S1 Supplementary Materials). The sample estimation of distance covariance for random vectors $X$, $Y$ is defined as:

$$\mathcal{V}_k^2(X, Y) = \|\phi_{X,Y}^k(t,s) - \phi_X^k(t)\phi_Y^k(s)\|_w^2, \tag{4}$$

and the sample estimation of DC is

$$DC(X, Y) = \mathcal{R}_k^2(X, Y] = \frac{\mathcal{V}_k^2(X, Y]}{\sqrt{\mathcal{V}_k^2(X, X]\mathcal{V}_k^2(Y, Y]}}. \tag{5}$$

It can be proved that under the independence hypothesis, as $m$, $n$ tend to infinity,

$$\mathcal{T}_k = \sqrt{\nu - 1} \cdot \frac{\mathcal{R}_k^2}{\sqrt{1 - (\mathcal{R}_k^2)^2}} \tag{6}$$

converges to $t$-distribution with $\frac{k(k-3)-2}{2}$ degrees of freedom. Thus, we can calculate the $p$-value easily for each hypothesis testing. We can see that the DC doesn't depend on the order of samples of both datasets from the derivation process (S1 Supplementary Materials).

**Select approximate observations for each gene.**   Naturally, multiple observations for each gene across both datasets are needed to estimate the empirical characteristic functions (S1 Supplementary Materials). However, we usually only have one observation for each gene across pairwise biological omics data. Profiling the given cells for multiple times to achieve this is unrealistic, since the cells are destroyed during the measurement process. Thus, it is impossible to perform the estimation directly. As we know, a set of genes usually tend to be highly correlated under the same condition due to the the modular organization of biological systems. In view of this, we introduce an approximate strategy to select $k$ highly correlated ones with it as alternative observations in a single dataset to overcome this issue. We applied this procedure to all the genes individually. The value of $k$ is set to 30 by default.

**The BH adjustment for controlling FDR.**   Since we repeat to calculate ADC for each gene, we expect to control the FDR, which is defined as the expectation of false discovery proportion (FDP):

$$\text{FDR} = \mathbb{E}[\text{FDP}] = \mathbb{E}\left[\frac{\#\{i : (X_i, Y_i) \text{ are independent and } i \in \hat{S}\}}{\#\{i : i \in \hat{S}\} \vee 1}\right], \tag{7}$$

where $X_i \in R^m$ and $Y_i \in R^n$, $i \in \{1, \ldots, p\}$, $\hat{S}$ is a subset of $\{1, \ldots, p\}$ denoting highly interrelatedd genes selected by ADC, $a \vee b = \max\{a, b\}$ and  denotes the size of a set. In the language of hypothesis testing, we are interested in the $m$ hypotheses $\{H_i: X_i \text{ and } Y_i \text{ are independent}\}$ and want to control FDR. For instance, if we control the FDR at 10% and select 100 pairs of genes, then at most ten of them are false.

The BH adjustment strategy is a classical method to control FDR for multiple hypothesis testing [24]. Consider multiple testing $H_1, H_2, \ldots, H_p$ with corresponding $p$-value $P_1, P_2, \ldots, P_p$, we order the $p$-value so that $P_{(1)} \leq P_{(2)} \ldots \leq P_{(p)}$. The adjusted $p$-value $Q_{(i)}$ for $P_{(i)}$ is calculated as follows:

$$Q_{(i)} = P_{(i)} \times \frac{p}{i}. \tag{8}$$

For a given FDR level $\alpha$, we call the $i$th testing is significant and only if $Q_{(i)} < \alpha$.

## Computational complexity of ADC

To rapidly select the $k$ approximate observations, ADC first constructs a Pearson correlation matrix for each dataset, and its computational complexity is $O(p^2 \max\{n, m\}]$. After that, ADC searches approximate observations for each gene based on the quicksort algorithm, and its total complexity for all genes is $O(p^2 \log p)$. Then, ADC calculates the DC for each gene pair across the two datasets, and its complexity is $O(k^3(k + \max\{n, m\}])$ (S1 Supplementary Materials). Since $k$ is a constant in ADC, the complexity of calculating the $p$-value of DC for all genes here is $O(p \max\{n, m\}]$. The BH adjustment is based on a sorting method, so its complexity is $O(p \log p)$. Taken together, the computational complexity of ADC is $O(p^2 \max\{n, m\}] + O(p^2 \log p) + O(p \max\{n, m\}] + O(p \log p) = O(p^2(\max\{n, m\} + \log p))$. Generally, the $\log p$ is far more less than $\max\{n, m\}$ for biological datasets, so the complexity can be reduced to $O(p^2(\max\{n, m\}]$. It is worth noting that in gene expression datasets, $p$, denoting the number of genes, is usually less than twenty thousand which will not increase so much, so the computational complexity of ADC is a linear complexity relating to the number of cells. Simulation experiments further confirmed this. Although the complexity is quadratic with $p$, the growth rate of

quadratic functions is very slow in the simulation study (Fig B(a) in S1 Supplementary Materials). All these results suggest that ADC is very efficient in handling large-scale datasets. ADC was implemented in Python and the package can be downloaded from https://github.com/zhanglabtools/ADC.

### Quantify the degree of similarity between two biological datasets

How to quantify the underlying similarity of datasets across different conditions, cell types, technologies and modalities is an important issue. For two single-cell omics data, we think that the expression patterns of genes in the same types of cells tends to be more similar. It is expected that ADC can detect more interrelated genes in the datasets with more common cell types. Thus, given a series of datasets, the number of genes selected by ADC could serve as a metric to measure the similarity between any two different datasets. The reciprocal of this number can be used to construct a distance matrix, and hierarchical clustering methods can be further applied it to build the hierarchical relation among these datasets. This could play key roles in many situations such as clustering cancer types with strong molecular similarity, measuring the consistency across technologies, exploring the degree of differentiation of progenitor cells, and mining the biological signals preserved across modalities.

## Results

### Simulation studies

In the simulation studies, we control the FDR with the BH adjustment method and verify the power of DC for selecting interrelated vectors. Here power means the probability of selecting genes exhibiting similar expression patterns. We considered four different simulation scenarios with 1000 vector pairs for each, half of them share similar structures. The dimensions of each pair of $X$ and $Y$ are $m = 3000$ and $n = 10000$, respectively. They are relatively large to the observation size 30, and the association of $X$ and $Y$ is built through the $l$ shared dimensions. The nominal FDR level in all the experiments are 20%.

**Scenario 1**. Draw $X = (x_1, \ldots, x_m)^T$ independently from a standard normal distribution. Let $y_i = sx_i$ $(i = 1, \ldots, l)$ and draw $y_i$ $(i = l + 1, \ldots, n)$ independently from a standard normal distribution, where $s \sim U(1, 5)$.

**Scenario 2**. Draw $X = (x_1, \ldots, x_m)^T$ independently from a standard normal distribution, after that 90% of the entries are replaced with zeros randomly. Let $y_i = sx_i$ $(i = 1, \ldots, l)$ and draw $y_i$ $(i = l + 1, \ldots, n)$ independently from a standard normal distribution, where $s \sim U(1, 5)$, after that 90% of the entries in $y_i$ $(i = l + 1, \ldots, n)$ are replaced with zeros randomly.

**Scenario 3**. Draw $X = (x_1, \ldots, x_m)^T$ independently from $U(0, 1)$. Let $y_i = \log(x_i)$ $(i = 1, \ldots, l)$ and draw $y_i$ $(i = l + 1, \ldots, n)$ independently from a standard normal distribution.

**Scenario 4**. Draw $X = (x_1, \ldots, x_m)^T$ independently from $U(0, 1)$, after that 90% of the entries are replaced with zeros randomly. Let $y_i = \log(x_i)$ $(i = 1, \ldots, l)$ and draw $y_i$ $(i = l + 1, \ldots, n)$ independently from a standard normal distribution, after that 90% of the entries in $y_i$ $(i = l + 1, \ldots, n)$ are replaced with zeros randomly.

Numerical results demonstrated that for all the four scenarios, FDR could be controlled well as expected with $l > 25$, and the power increased quickly with the $l$ growing (Fig 2). Further, the FDR is around 10%, which is only a half of the nominal level. Thus, this method is very conservative and we could expect much lower FDR in real applications. Moreover, even the two vectors only have $l = 100$ related dimensions, which is very small compared with the lengths of the two vectors, the power will exceed 90% or 80% for the vectors with linear or nonlinear dependence structures, respectively (Fig 2A and 2C), indicating DC is very sensitive to capture tiny similarities. More importantly, sparseness does not hinder the power of DC even

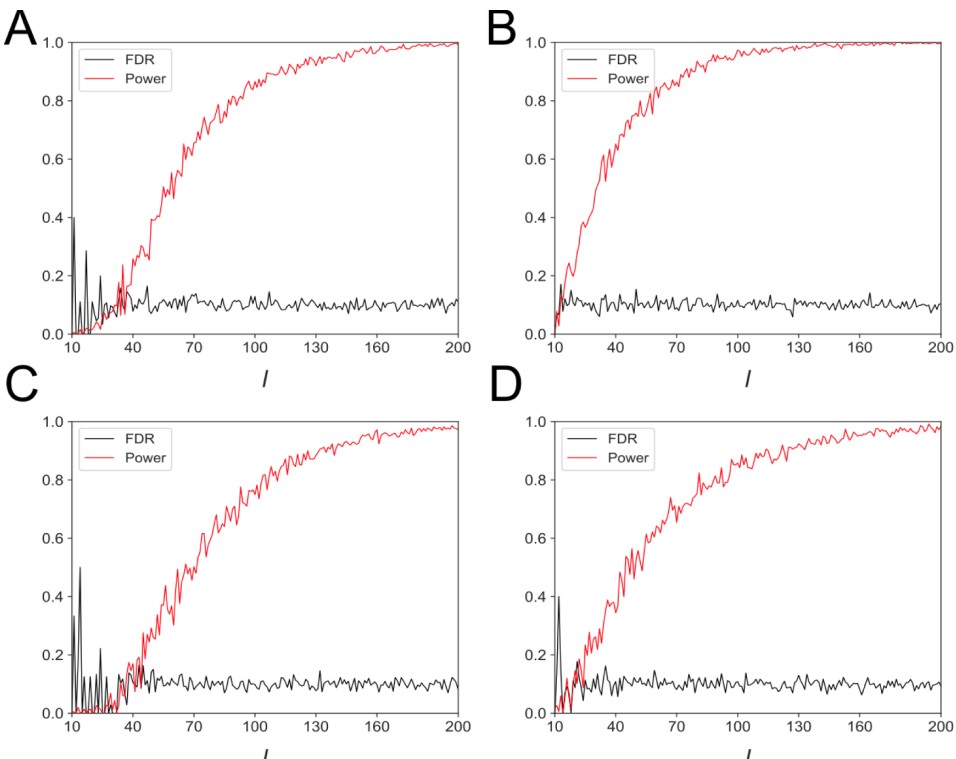

**Fig 2. Simulation experiments on DC combined with the BH method in terms of Power and FDR (the target level is 20%).** (A) Each pair of vectors are dense, and $k$ dimensions are shared with a linear transform. (B) Each pair of vectors are sparse with 90% zero entries and $k$ dimensions are shared with a linear transform. (C) Each pair of vectors are dense and $k$ dimensions are shared with a non-linear transform. (D) Each pair of vectors are sparse with 90% zero entries and $k$ dimensions are shared with a non-linear transform.

if zero-entries of the data is up to 90% (Fig 2C and 2D). These results hold true even the random vectors were generated from a much more complicated distribution (Fig A in S1 Supplementary Materials). Therefore, DC combined with the BH adjustment method is a powerful method to capture the dependent structure in data with high power and stable FDR control. When applying to biological data, we expect to select genes with similar expression (or regulatory) patterns. The length of the gene vector is the number of cells, we believe that even if genes are interrelated in very few cells across two datasets, DC can still accurately identify them.

Next, we used the splatter R package [25] to simulate three scRNA-seq datasets with four cell types (Fig 3A). As expected, the results of ADC is consistent with the data structure (Fig 3B), and it selected most correlated genes between Data 1 and Data 3. Although there are only 40% cells overlapped between Data 1 and Data 2, ADC accurately captured the weak associations and revealed fewer correlated genes than those selected between Data 2 and Data 3 as expected. We also test the influence of hyper-parameter $k$ by applying ADC to Data 1 and Data 2. Here, we selected the top 100 highly interrelated gene under each $k$ varying from 20 to 40. For each pair of $k$, the number of overlapped genes was calculated. In average, we got almost 93 genes, that suggests ADC is a robust algorithm considering the selection of $k$ (Fig B (c) in S1 Supplementary Materials).

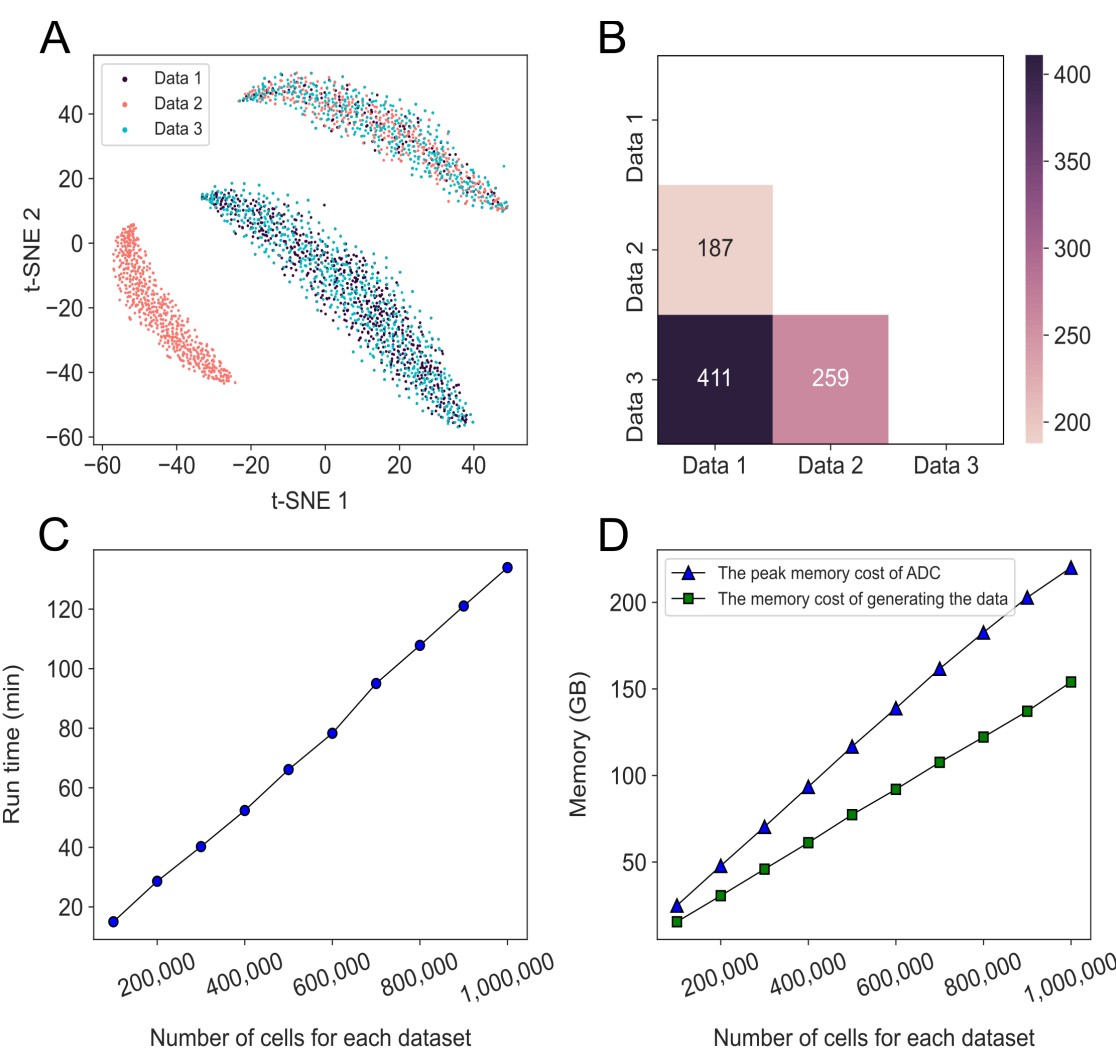

**Fig 3. Performance of ADC on three simulated datasets.** (A) The t-SNE plot of these datasets with three cell types in total. 80% of cells in Data 1 and Data 3 are of the same type, 60% of cells in Data 2 and Data 3 are of the same type, and 40% of cells in Data 1 and Data 2 are of the same type. Each dataset contains 1000 cells and 5000 genes. (B) Heat map of the highly interrelated genes selected by ADC across Data 1, Data 2 and Data 3 (FDR = 0.05). (C and D) Running time and peak cost of ADC with two datasets with 1 million cells and 10 thousand genes each in no more than 135 mins and under 225 GB of RAM. Each entry of the datasets was generated with a random variable which obeys uniform distribution. GB indicates the GigaByte.

Moreover, ADC could handle two datasets with ten thousand cells each in about 15 minutes. The running time increased linearly in term of the number of cells (Fig 3C), which confirms our derivation on the computational complexity. The peak memory cost of ADC is less than twice of generating the data, indicating ADC consumes less memory than copying the data (Fig 3D). All results suggest that ADC is a powerful method and applicable to large-scale datasets.

## Highly interrelated genes across-cell types reveal cancer similarities

We applied ADC to the gene expression data of 21 different cancers and selected highly interrelated genes across each pair of cancers (Fig 4A). The number of highly interrelated genes between each pair of cancers clearly reflects the degree of similarities between different cancers

(Fig 4A). Hierarchical clustering based on it demonstrates that cancers generated in similar tissues are clustered together, such as GBMLGG and LGG, KIPC and KIRP, COAD and COAD-READ, SRAD and STES. Further, the two cancer types BRCA and OV occurring most frequently in women demonstrate great similarity. Besides, we also found a cluster of two squamous carcinoma, HNSC and CESC, which have been shown to have strong molecular similarity [26]. All these observations indicate that ADC could gain deep insights into the common characteristic of cancers.

Clearly, the enriched biological functions of the genes selected from two pairs of cancers implicate to key cancer hallmarks [27] such as cell-cell adhesion, T cell proliferation, immune system process and response to interferon-$\gamma$, suggesting that these selected genes are indeed biologically relevant (Fig 4B and 4D). We also constructed gene functional networks by Gene-MANIA [28] with the physical and genetic interactions of these genes (Fig 4C and 4E). The hub gene APPL1 of the first network is related with cell cycle [29]. Moreover, it is an immune cell enhanced gene and the protein encoded by this gene has been shown to be involved in the regulation of cell proliferation [30]. In the second functional network, the hub gene EMSY is an oncogene linking the BRCA2 pathway to sporadic breast and ovarian cancer [31] and is involved in several cancer related biological processes like DNA damage, DNA repair, transcription and transcription regulation. The enriched functional terms of other cancer pairs also reflects cancer hallmarks (Fig C in S1 Supplementary Materials). These results shows that ADC indeed could find many strongly associated genes of two cancers.

## Highly interrelated genes across cell types reveal their hierarchical lineage

Here we applied ADC to six distinct hematopoietic stem and progenitors from mouse bone marrow at single cell level (Fig 5A). The numbers of highly interrelated genes of any two cell types clearly reveal their lineage structure (Fig 5B). As expected, HSC giving rise to all hematopoietic lineages, shows distinct differences with other cell types at the end of differentiation according to the selected genes based on ADC (Fig 5B). For HSC, ADC selected the most correlated genes with LMPP than any other cell types, we speculated that LMPP could be directly derived from HSC, which was confirmed by a recent study [32]. CMP is an early myeloid progenitor cell which differentiates to GMP and MEP, but it shows much more similarities with its progenitor MPP. Moreover, we found that CMP have more similarities with GMP compared with MEP based on the number of selected genes (182 vs 173). This was first confirmed by visualization of these cells (Fig D(a) in S1 Supplementary Materials), where CMP clearly has more cells overlapped with GMP. We further applied an unsupervised clustering method leiden [33] to identify 13 clusters (Fig D(b) in S1 Supplementary Materials), and CMP shows similar cell distribution with GMP compared to MEP (Fig 5C), and the cells overlapped between these two cell types are far more than that between CMP and MEP (Fig 5D). Further, the CMP and GMP belong to myeloid cells while MEP belong to the erythroid cells, and a study also shows that CMP and GMP preferentially differentiate into proangiogenic cells compared with MEP [34]. Thus, CMP and GMP should be more similar at the gene expression level, and the same result has also been observed from the human hematopoietic cell data (Fig 5B). All these results indicate that ADC can capture the true genetic similarities and infer cell lineage structure using the sparse scRNA-seq data.

## Highly interrelated genes across technologies illustrate their inherent difference

Here we applied ADC to five scRNA-seq datasets of pancreatic islet cells profiled by indrop, CEL-Seq2, 10X, CEL-Seq, Smart-seq2, respectively. Obviously, common cell types like alpha,

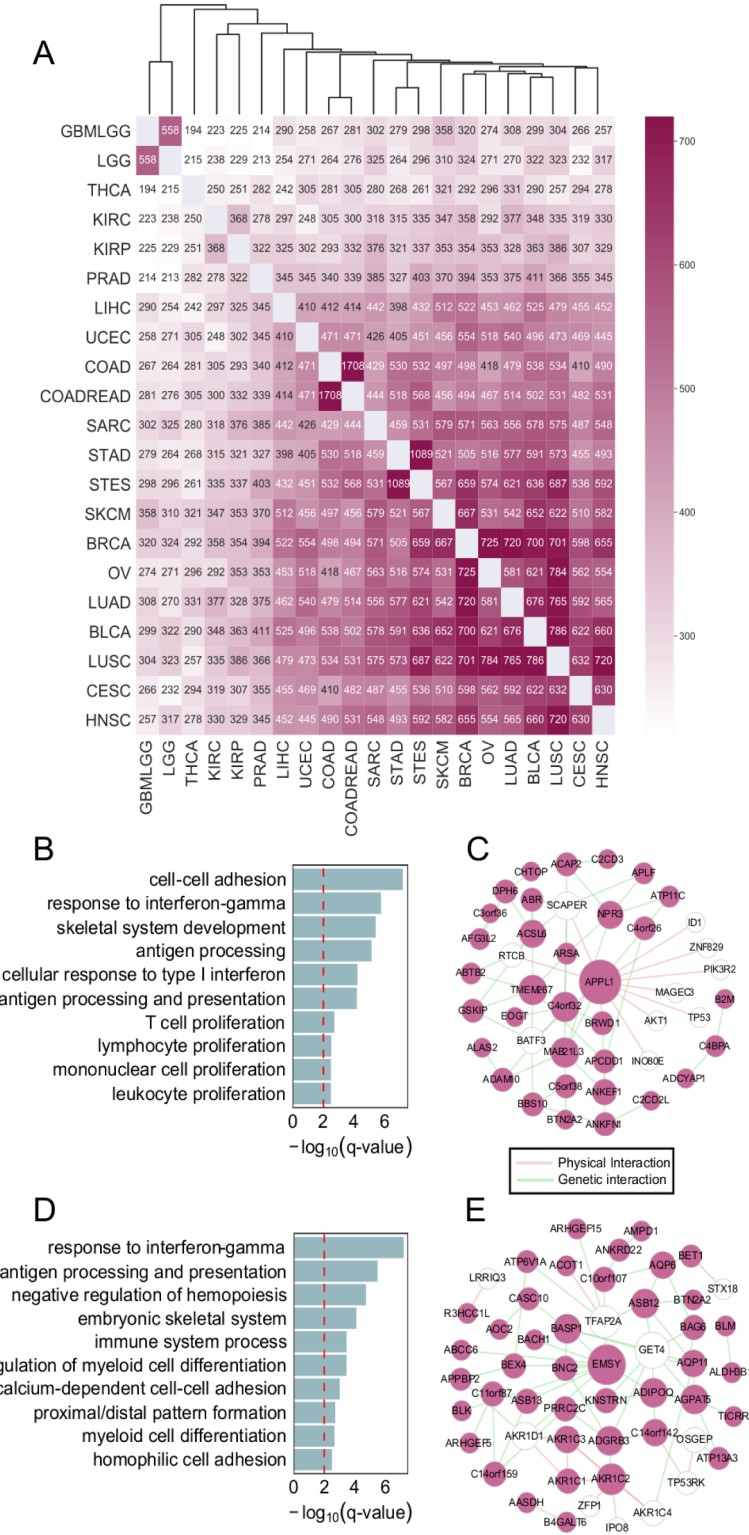

**Fig 4. Highly interrelated genes and their biological functions among 21 cancers.** (A) Heatmap of the number of highly interrelated genes selected by ADC between each pair of cancers (FDR = 0.05). Hierarchical clustering was performed with the reciprocal value of the number. (B and D) The top ten enriched functional terms of these selected genes between HNSC and CESC (B), BRCA and OV (D) respectively. (C and E) The gene network constructed with GeneMANIA using the selected genes between HNSC and CESC (C), BRCA and OV (E) respectively.

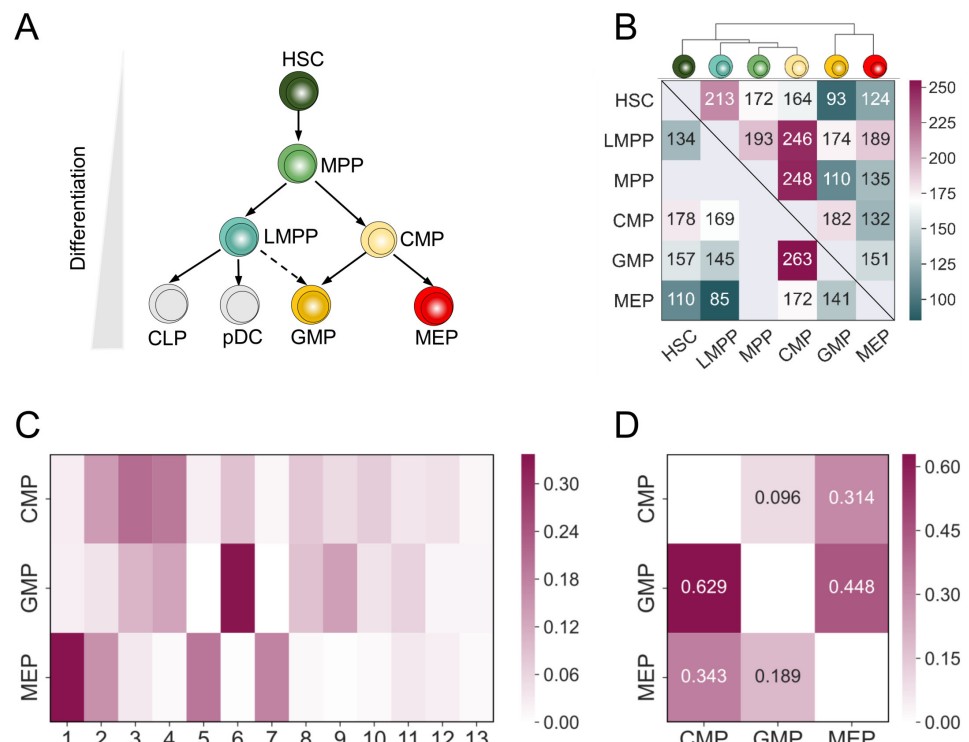

**Fig 5. Highly interrelated genes selected by ADC between different cell types along the hematopoietic cell lineage.**
(A) A schematic of mouse hematopoietic differentiation. The cells in gray color cell type were not present in our datasets. (B) Heat map shows the number of highly interrelated genes selected by ADC cross six cell types (FDR = 0.10). The upper triangular is the result of mouse hematopoietic cells while the lower triangular is the result of human hematopoietic cell data downloaded from GEO with accession code GSE117498. Unsupervised hierarchical clustering analysis was performed with the reciprocal value of the number. (C) Confusion matrix of data-driven clusters representing the percent frequency distribution of immunophenotypically defined cell types. (D) Heat map shows the dissimilarity of the distributions of the three cell types using the Jensen-Shannon (JS) divergence.

beta, ductal, acinar are shared by these five datasets. Conceptually, highly variables genes (HVGs) among data should strongly contribute to cell-to-cell variation within relatively homogeneous cell populations [35]. That is to say, HVGs could depict the cell characteristics well. However, we find that the top HVGs overlapped among these datasets cannot reflect the similarities between these technologies (Fig 6A). The two linear amplification methods CEL-seq and CEL-seq2 are separated in two different clusters irrationally [36, 37]. While the two droplet-based methods, 10X and indrop, which can profile a mass of cells and have a relatively low resolutio [38], also shows less similarity than that between 10X and CEL-seq2. Even if we increased the number of HVGs (Fig E(a-d) in S1 Supplementary Materials), the results still failed to reveal the relative differences of these technologies. Strikingly, ADC could identify a number of correlated genes across-technology data, and reveal the technology similarity across them (Fig 6B). Specifically, CEL-Seq showed the most similarity with its modified version CEL-Seq2, 10X and indrop were clustered together while other three methods which profiled genes with a higher resolution showed distinct similarities.

## Similarly correlated genes across modalities reveal biological relevant genes

Next we applied ADC to scATAC-seq (using gene activity matrix [39]) and scRNA-seq data-sets of human peripheral blood mononuclear cells (PBMC) with 8728 cells and 2638 cells

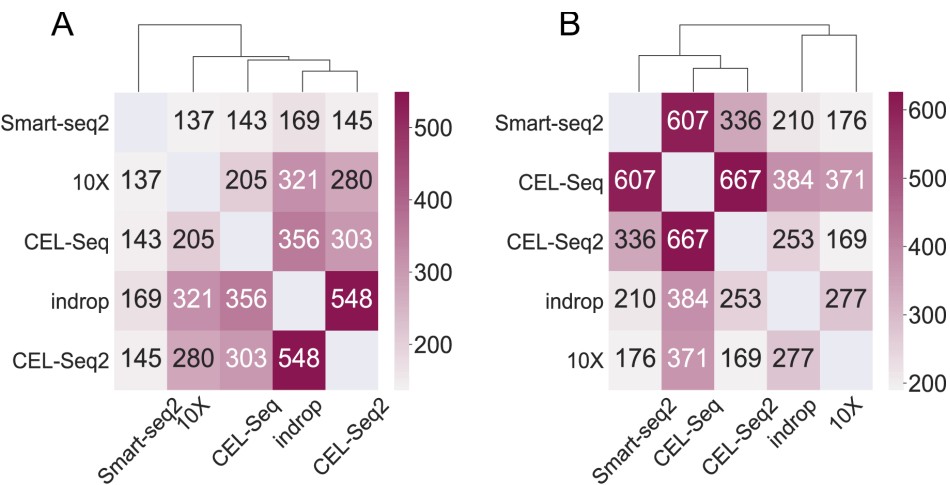

**Fig 6. (A) Number of overlapping genes among the top 1000 HVGs of the data from five different technologies, and (B) Highly interrelated gene selected by ADC between each pair of these datasets (FDR = 0.01).** Unsupervised hierarchical clustering was performed with the reciprocal value of the number in both situations.

respectively to select correlated genes across them (Fig 7A). ADC revealed 195 highly related genes, which are involved in PBMC related biological process like interferon-gamma production, leukocyte differentiation, antigen receptor-mediated signaling pathway (Fig 7B) [40, 41]. The hub gene of the network, GNAI2, which regulates the entrance of B Lymphocytes into lymph nodes and B cell motility within lymph node follicles [42]. Besides, another hub genes FMNL2, is also enriched in monocytes [30]. All these observations demonstrate that although scATAC-seq and scRNA-seq data are different types, we can still explore molecular similarities across these different modalities.

## Discussion

With the rapid development of high-throughput sequencing technologies (e.g. RNA-seq, scRNA-seq, scATAC-seq), a huge number of biological datasets under different conditions have been profiled. Here we proposed ADC to analyze the pairwise data generated from different tissues, conditions or technologies with high efficiency. To our knowledge, this is the first method to identify highly interrelated genes across different biological datasets.

ADC derived from DC measures the correlation of random variables in arbitrary dimensions. As we shown in the simulation studies, combined with the BH adjustment method, DC could accurately detect associated variables with high power and low FDR, even there are only few related dimensions. Moreover, the number of selected genes can reflect the degree of similarity between datasets. ADC is applicable to massive datasets with millions of samples in several hours. Extensive tests on four real applications demonstrate its effectiveness. Also, ADC can find tissue-specific genes which are interrelated across human and mouse, where these genes capture conserved functions across them (Fig F in S1 Supplementary Materials). Thus, ADC is expected to be applied to diverse biological datasets under different conditions.

The number $k$ of selected highly correlated genes in each dataset is an important parameter which could affect the gene sets for DC calculation. We have empirically demonstrated that the default one work well in the simulation studies and biological applications. Alternatively, we could provide a more robust version by leveraging the interrelated genes selected by ADC with different $k$s. Specifically, we detected the interrelated gene under each $k$ varying from 20

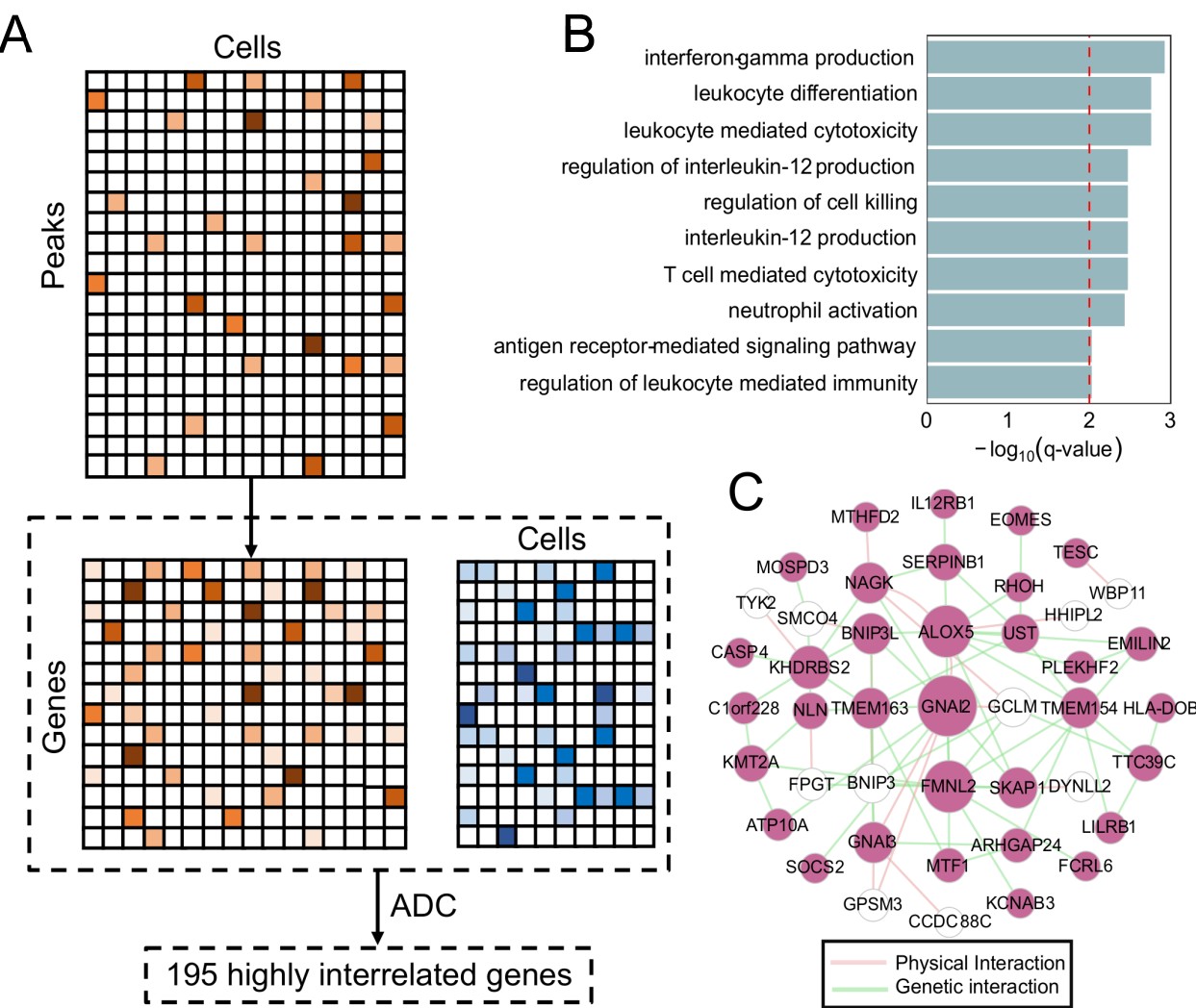

**Fig 7. Biological functions and network analysis of highly interrelated genes selected by ADC across modalities.** (A) Schematic diagram shows the work flow of performing ADC on the scRNA-seq data and scATAC-seq data (FDR = 0.20). We first converted the scATAC-seq data to a predicted gene expression matrix. Specifically, we constructed a "gene activity matrix" from scATAC-seq dataset by utilizing the reads at gene body and 2kb upstream, then ADC was applied to a pseudo gene expression data and a real one. (B) The top enriched functional terms of the genes selected by ADC. (C) The gene network constructed with GeneMANIA using the top genes selected by ADC.

to 40, and we kept genes which were considered as interrelated ones under 21 different *k*s. The results are very consistent with the original ones (Fig 6B and Fig E(e) in S1 Supplementary Materials).

Recently, with the improvement in high-throughput single cell technologies, different experiments such as chromatin accessibility [43], DNA methylation [44], RNA modification [45] and gene expression datasets at single cell level are increasing rapidly (Fig G in S1 Supplementary Materials). Many methods have been put forward to process these data [46–48]. Most importantly, integration analysis to transfer knowledge from one dataset to another has became increasingly important [49]. For example, Haghverdi *et al.* [50] used mutual nearest neighborhoods (MNN) on the orginal data space to find the same cell types. Stuart *et al.* [23] improved this methods by performing MNN on the reduced dimension learned by canonical correlation analysis. However, these methods only focus on the integration problem itself but

ignore whether these datasets can be integrated or not, which may lead to "overcorrection" [51]. Moreover, they may fail to consider the order of the integration. While ADC can provide the order of integration based on the number of selected genes, which can help to improve the integration result. Also, from the perspective of knowledge transfer, given a newly generated data with limited prior knowledge, we would like to select a most correlated dataset which has been fully explored as a reference. ADC is a suitable tool under this circumstance since it can quantitatively measure the degree of similarity between two datasets. Furthermore, ADC can also uncover interrelated variables at molecular level, which may increase our insights into a new dataset. Because ADC has no assumption, it can be directly applied to other biological datasets without any modification. ADC is a valid data-driven method which doesn't employ any external information now. It will be an interesting direction to consider external knowledge (e.g., gene ontology) to improve it in future.

## Supporting information

**S1 Supplementary Materials. Supplementary figures, tables and analysis details.** Further detailed descriptions of the principle of ADC, the computational complexity of ADC, and more information about the methods and process of data analysis for single-cell RNA-seq data. **Fig A**. **Simulation experiments on DC combined with the BH method in terms of Power and FDR (target is 20%)**. We generated each pair of variables with 3000 and 10,000 dimensions, respectively. Every non-zero entry of the variables was sampled from a beta(2,4) distribution. (a) Each pair of vectors are dense and $k$ dimensions are shared with a linear transform. (b) Each pair of vectors are sparse with 90% zero entries and $k$ dimensions are shared with a linear transform. (c) Each pair of vectors are dense and $k$ dimensions are shared with a log transform. (d) Each pair of vectors are sparse with 90% zero entries and $k$ dimensions are shared with a log transform. **Fig B**. **Performance of ADC on simulated datasets**. (a and b) Running time and peak menmory cost of ADC with two datasets with 10 thounsand cells and different number of genes. GB indicates the GigaByte. (c) We applied ADC to data1 and data2 generated by splatter (Fig 3A) under each $k$ from 20 to 40. For each $k$, we selected top 100 interrelated genes and calculated the number of overlapped top genes for each pair of $k$. The result is showed in the boxplot. **Fig C**. **Functional enrichments of selected genes between five pairs of cancers**. (a) BLCA and LUSC, (b) GBMLGG and LGG, (c) KIRC and KIRP, (d) STAD and STES, and (e) COAD and COADRED. **Fig D**. **PCA plots of hematopoietic stem cells CMP, GMP and MEP**. (a) The cells are colored by the cell types annotated by the combination of molecular surface markers. (b) The cells are colored by the cell types annotated by an unsupervised clustering method Leiden. **Fig E**. **Numbers of selected genes across five technologies for the data with different number highly variable genes**. (a) top 2000 genes, (b) top 3000 genes, (c) top 4000 genes, and (d) top 5000 genes. Unsupervised hierarchical clustering analysis is performed with the reciprocal value of the numbers. **Fig F**. **Enriched GO terms of highly interrelated genes between human and mouse done by Metascape**. **Fig G**. **The number of scRNA-seq datasets per month from January 2013 to July 2020**. The statistics was obtained from NCBI Gene Expression Omnibus with key words: "single cell RNA seq" or "single cell transcriptome" or "single cell gene expression". **Table A**. **Number of samples of 21 types of cancer**. 17 out of 38 types of cancer in TCGA were excluded in our study due to limited numbers of samples. **Table B**. **Number of cells of six hematopoietic cell types**. **Table C**. **Number of pancreatic islet cells from five different technologies**. **Table D**. **Numbers of PBMC cells from two different sequencing methods**.
(PDF)

## Author Contributions

**Conceptualization:** Qunlun Shen, Shihua Zhang.

**Funding acquisition:** Shihua Zhang.

**Investigation:** Qunlun Shen, Shihua Zhang.

**Methodology:** Qunlun Shen, Shihua Zhang.

**Project administration:** Shihua Zhang.

**Resources:** Qunlun Shen.

**Software:** Qunlun Shen.

**Supervision:** Shihua Zhang.

**Validation:** Qunlun Shen, Shihua Zhang.

**Visualization:** Qunlun Shen.

**Writing – original draft:** Qunlun Shen.

**Writing – review & editing:** Shihua Zhang.

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
