## [Decision Letter · Decision Letter 0]

9 Jun 2021

Dear Dr. Zhang,

Thank you very much for submitting your manuscript "Approximate distance correlation for selecting similarly distributed genes across datasets" for consideration at PLOS Computational Biology.

As with all papers reviewed by the journal, your manuscript was reviewed by members of the editorial board and by several independent reviewers. In light of the reviews (below this email), we would like to invite the resubmission of a significantly-revised version that thoroughly addresses the reviewers' comments. In particular, as pointed out by multiple reviewers, you should significantly improve the rigor of the method evaluation and data interpretation and further demonstrate the advantages over existing methods (e.g., for data integration application). In addition, please make the source code associated with the proposed method publicly available.

We cannot make any decision about publication until we have seen the revised manuscript and your response to the reviewers' comments. Your revised manuscript is also likely to be sent to reviewers for further evaluation.

Sincerely,

Mingyao Li

Associate Editor

PLOS Computational Biology

Jian Ma

Deputy Editor

PLOS Computational Biology

Reviewer's Responses to Questions

**Comments to the Authors:**

Reviewer #1: In this study, the authors developed a method to study the gene expression conservation of each gene between different datasets based on distance correlation using co-expressed genes. They showed that their method can identify conserved genes from bulk RNA-seq, scRNA-seq, and scATAC-seq datasets. Although the results are compelling, the paper can be further improved. See detailed comments below.

1. The authors select k=30 highly correlated genes in each dataset to calculate the distance correlation. How the selection of k affects the result is not clear. Why selecting k=30 as the default value has to be justified.

2. How batch effects affect the result is not examined. The authors showed in Figure 6 that their method can capture more similar genes between sequencing platforms that are similar. It shows batch effects (here platform effects) are clearly affecting the results.

3. The proposed method aims to reveal genes that have similar expression patterns between two datasets. Can it be generalized to more than two datasets? For example, can it be applied to reveal common genes across all cancer samples showed in Figure 4?

4. Although the method is designed to reveal genes that are similar between datasets, can it reveal genes that are most diverse (not similar) between datasets?

5. For the hematopoietic single-cell RNA-seq analysis in Figure 5B, the author only showed results for CMP, GMP, and MEP for human bone marrow. How about the results from other cell types? Is it also consistent with the mouse data?

6. Another interesting question the authors haven’t explore is cross-species conservation. Can the proposed method reveal genes that have similar expression patterns between human and mouse?

7. For Figure 7, the authors studied the similar genes between PBMC scRNA-seq and scATAC-seq datasets. Recently, multi-modal scRNA-seq+scATAC-seq datasets are also available for PBMC which can be obtained from the 10X genomic website. It will be interesting to see if the genes identified by the proposed method between scRNA-seq and scATAC-seq are similar to those identified using simple correlation metrics based on the multi-modal datasets.

8. The application of the proposed method to data integration is not convincing. The state-of-the-art methods for single-cell data integration such as Harmony are not affected by the order of sample in integration. Also, the example is too simple. If the authors want to show the usage of their method in integration, they may want to provide a more comprehensive analysis.

Reviewer #2: In this paper, the authors propose to use the Approximate Distance Correlation (ADC) to identify similarly distributed genes between two biological datasets, and subsequently to use the identified genes to measure the similarity between these two datasets. For each gene, they first identify its k most correlated genes within each dataset, and then they use the measurements of these two sets of k genes in their corresponding datasets to calculate the distance correlation (DC), which they define as the ADC.

Although the paper is overall clearly written, we have several major concerns about the validity of the methodology. In our opinion, these concerns, if not addressed, will preclude ADC to be a useful data analysis tool.

1. Our first concern is the motivation of ADC. If the goal is to identify genes with similar distributions in two datasets, then what is the definition of “distributions.” To us, this is the key and should be clarified in the beginning.

2. We have a major concern about applying the distance correlation to the manually selected k genes in each dataset. In Section “Materials and methods”---"Select approximate observation for each gene," distance correlation requires (X_i, Y_i) to be a true pair, and {(X_i, Y_i)} to be iid. Here, however, (X_i, Y_i) is not from the same gene and thus not naturally paired, and {(X_i, Y_i)} are neither independent (they are manually selected, with high correlation) nor identically distributed (they are from different genes). Hence, the basic assumptions of the distance correlation are violated.

3. Due to the fact that there exists only one observation for each gene, we suggest that the authors view the biological problem on a metagene level as a remedy. To be more specific, authors can first cluster genes into metagenes in the original datasets, and then they can calculate the distance correlation for each metagene. In this way, replicates (X_i, Y_i) are naturally paired for the i-th gene in metagene.

4. If the authors do want to achieve the single-gene resolution, to remedy the violation of the assumptions required by the distance correlation, the authors may consider taking the advantage of gene orthology information instead of using the same data twice to select the top k correlated genes for every gene. At least the authors must make sure that (X_i, Y_i) is paired by external information, not the data to be used to calculate the distance correlation.

5. There's little discussion on the choice of parameter k (number of selected highly correlated genes in each dataset). It affects the sets of genes for distance correlation calculation, as well as the asymptotic distribution. Instead of simply proposing a default value of k, the authors should discuss the effect of k and provide guidance on choosing k.

6. In Fig 6A, we see that results are far from satisfactory in the overlap of top HVGs. We wonder whether this is a fair comparison. Are the selected top HVGs restricted to the common genes shared between datasets?

7. The authors should apply more direct validation other than comparing the number of similarly expressed genes and the data structure. For example, to highlight that the algorithm can detect similarly distributed genes across datasets, the authors can directly compare the overall expression pattern of the same genes with high ADC scores in both datasets.

Reviewer #3: In this manuscript, the authors develop a statistical method, namely Approximate Distance Correlation (ADC) for detecting genes with similar expression patterns across multiple samples or datasets. ADC is an alternative to DC, where multiple samples are needed for calculation. Instead of multiple samples, ADC utilizes the genes that are highly correlated with a gene, as alternative observations. The authors formulate their solutions and show its effectiveness with multiple case studies.

Although the idea seems interesting and the case study results are impressive, there are some problems with the manuscript, as pointed out below:

Major Comments:

1. The most important issue about this method is using the correlated genes as alternative observations for a gene. Authors tell that this is necessary because often the experiment cannot be repeated for the same sample, noting that “Profiling the given cells for multiple times to achieve this is unrealistic, since cells are destroyed during the measurement process.” However, in the same manner, using correlated genes as alternative observations is similarly unrealistic, since almost every gene has its own distinct expression pattern. In this sense, I think it is highly questionable to use correlated genes for this purpose, instead of relying on multiple cells, which are usually in the same biological state. I think the approach followed here, using ADC instead of DC, is only diverting the problem of reproducibility on a different aspect, instead of solving it.

2. In this context, the authors should compare their method to using Distance Correlation (DC) for each benchmarking and case study. Does their method (ADC) outperform DC, by using the same set of samples? Since ADC depends on correlation, which requires multiple samples or cells, DC can be computed and used for every test.

3. Page 5, Line 117: “Profiling the given cells for multiple times to achieve this is unrealistic. Since cells are

destroyed during the measurement process” should be single sentence “Profiling the given cells for multiple times to achieve this is unrealistic, since since the cells are destroyed during the measurement process.”

4. Page 7, Line 176: What do m and n denote? Genes and cells?

5. Page 8, Line 216: “Although there are only 40% cells overlapped between Data 1 and Data 3” contradicts with the Fig. 3 legend: “40% of cells in Data 1 and Data 2 are of the same type.” Is it Data 2 or Data 3?

6. Page 9, Figure 3 legend: FDR is chosen 0.20? Why so large cutoff? Generally FDR is chosen 0.01 or 0.05, very rarely 0.1 at most.

7. Page 9: The authors do some GO annotation for similar genes and find cancer hallmark terms as a result. Using DC or just overlapping highly expressed genes can also give the same results. Is ADC superior in any sense? And just finding halmark terms doesn’t mean ADC could gain deep insights, since this is nothing new. It is just some kind of validation.

8. Page 10: Figure 4 C and E. In the heatmap, hundreds of genes are reported but very small number of genes are shown in C and E. How are those genes selected? Why is particularly special about those genes? What are the node sizes proportional to? Why are some genes colored pink and others white?

9. Page 11. Figure 5A. What are those cell types (HSC, MPP, etc.)? Since not every reader is a hematologist, these acronyms should be defined in the figure caption.

10. Page 12, Line 287 and Fig. 6: “Thus, top HVGs of the five datasets are supposed to have many overlaps since they were all profiled about the similar cell types. However, we only found a few HVGs overlapped among these datasets (Fig. 6A).” The numbers in the heatmaps in Fig 6A and 6B are not comparable. If you start with 1000 genes for an analysis and 20,000 genes for the other, you will obviously get different numbers in total. The actual numbers are not comparable between two. Only clustering can be compared.

11. Page 13, Figure 7C: Similar issue. Like Fig 4 C and E, how are those genes selected? Why is particularly special about those genes? What are the node sizes proportional to? Why are some genes colored pink and others white?

12. Although the authors developed a method, they did not provide the computational implementation of their framework in any platform. In this context, this study has little benefit for the scientific community. The authors must compile a software package in a widely used open-source programming language (such as R) not a commercial software language (such as Matlab), so that any researcher interested in this method and can freely download and apply this method on their own data. The software must be deposited in a publicly accessible repository such as GitHub and properly documented with code comments, package reference and tutorials. Any researcher should be able to download and apply it in a couple hours without need to intensive effort.

13. In addition to publishing the software package for the framework, the authors should compile a separate package for just re-producing all the results presented in this manuscript. The authors must provide all the code, scripts and documentation in addition to interim data to replicate the results presented. If the public datasets that are used is reasonable size, the authors should also deposit it in a repository for easy access. If some of the public data is very large (larger than > 100GB), the authors should provide links to download the data with information about how to process it.

**Have the authors made all data and (if applicable) computational code underlying the findings in their manuscript fully available?**

Reviewer #1: **No: **The computational code for the proposed method is not available.

Reviewer #2: None

Reviewer #3: **No: **Although the authors developed a method, they did not provide the computational implementation of their framework in any platform. In this context, this study has little benefit for the scientific community.

PLOS authors have the option to publish the peer review history of their article (what does this mean?). If published, this will include your full peer review and any attached files.

Reviewer #1: No

Reviewer #2: No

Reviewer #3: No
---

## [Decision Letter · Decision Letter 1]

31 Aug 2021

Dear Dr. Zhang,

Thank you very much for submitting your manuscript "Approximate distance correlation for selecting similarly distributed genes across datasets" for consideration at PLOS Computational Biology.

As with all papers reviewed by the journal, your manuscript was reviewed by members of the editorial board and by several independent reviewers. In light of the reviews (below this email), we would like to invite the resubmission of a significantly-revised version that addresses Reviewer #2's comments.

We cannot make any decision about publication until we have seen the revised manuscript and your response to the reviewers' comments. Your revised manuscript is also likely to be sent to reviewers for further evaluation.

Sincerely,

Mingyao Li

Associate Editor

PLOS Computational Biology

Jian Ma

Deputy Editor

PLOS Computational Biology

Reviewer's Responses to Questions

**Comments to the Authors:**

Reviewer #1: The authors have addressed all the comments.

Reviewer #2: The authors did not address the major concerns raised by Reviewers 2 and 3 regarding the validity of the proposed method ADC. Most importantly, the biological meanings of the genes selected by ADC are hard to interpret. Moreover, we do not agree with the authors that these genes should be referred to as "similarly distributed genes." We did a simulation study to show that a gene with the same distribution in two batches of cells cannot be captured by ADC. Hence, we are not convinced that ADC has a solid probability foundation or a biological relevance.

Below please find our detailed comments.

1. The biological meanings are unclear for the "similarly distributed genes" identified by ADC. The current ADC implementation tries to do the following. Take k = 4 as a toy example; for every tested gene, it selects (k-1) top correlated genes in both datasets, which may be {A, B, C} in dataset 1, and {D, E, F} in dataset 2; ADC will identify this tested gene as a "simiarly distributed gene" if and only if {tested gene, A, B, C} and {tested gene, D, E, F} are "correlated" in the sense of distance correlation (which roughly means that the 6 pairwise distances among {tested gene, A, B, C} and the 6 pairwise distances among {tested gene, D, E, F} are correlated). However, there is no guaranteed correspondence between {A, B, C} and {D, E, F}, and thus the distance correlation between {tested gene, A, B, C} and {tested gene, D, E, F} is biologically meaningless. Also, how does this correlation inform whether the tested gene is "similarly distributed" or not? Please see point 2 below.

2. There is a gap between what ADC claims to identify and what ADC actually identifies. What ADC claims to identify are the genes that have similar distributions in two datasets, where they have m and n observations, respectively. ADC aims to perform the following hypothesis testing: for every tested gene, X \\in R^m, Y \\in R^n are 2 random vectors, representing that gene's count vectors in m and n cells, respectively; H_0: X and Y are independent vs. H_a: X and Y are not independent" (Page 5, Section "DC"). While ADC claims/aims to identify those genes that reject H_0, what ADC actually does is different (please refer to our comment 1 that ADC actually identifies those genes whose top correlated genes are "correlated" between the two datasets). In the claim, X and Y should both correspond to the tested gene; however, in the actual implmentation, X refers a not-well-defined random vector whose four observations are {tested gene, A, B, C}, while Y refers a not-well-defined random vector whose four observations are {tested gene, D, E, F}. Given that A, B, and C are selected based on their high correlations with the tested gene in the first data set, it is impossible to assume that {tested gene, A, B, C} are i.i.d. observations of X. Hence, the actual implmentation of ADC is invalid in the probability sense.

We design the following simulation to demonstrate this gap:

Generate X1, X2, ..., Xm, Y1, Y2, ..., Yn ~ N(0, I_p) i.i.d.

Denote X = [X1, X2, ..., Xm], Y = [Y1, Y2, ..., Yn], where rows correspond to the p genes, and the m and n columns correspond to the m and n cells in two datasets.

Run the python code provided by the authors.

In the result, ADC does not identify any similar genes (FDR = 0.05).

This result is expected because for every tested gene, the top k correlated genes in X and Y are independently simulated and thus not correlated, so no gene should be identified by the ADC method as implemented. ("the actual goal")

However, by design, every gene is simulated from the same standard Gaussian distribution in the two datasets, so it perfectly satisfies the definition of "similarly distributed gene" and thus should be identified. ("the ideal goal")

Therefore, we conclude that the actual goal of ADC is different from its ideal goal.

3. To re-formulate ADC as a valid method, we deem it necessary for the authors to accurately and formally define "distribution" and "similar genes" using probability notations.

4. A potential fix: for a tested gene, if its top k correlated genes in the two datasets are paired by external information (e.g., gene orthology), shared, or naturally paired, the ADC result may carry more biological meaning.

Reviewer #3: My comments were addressed. The work is ready for publication.

**Have the authors made all data and (if applicable) computational code underlying the findings in their manuscript fully available?**

Reviewer #1: Yes

Reviewer #2: None

Reviewer #3: Yes

PLOS authors have the option to publish the peer review history of their article (what does this mean?). If published, this will include your full peer review and any attached files.

Reviewer #1: No

Reviewer #2: No

Reviewer #3: No
---

## [Decision Letter · Decision Letter 2]

10 Oct 2021

Dear Dr. Zhang,

We are pleased to inform you that your manuscript 'Approximate distance correlation for selecting highly interrelated genes across datasets' has been provisionally accepted for publication in PLOS Computational Biology.

Best regards,

Mingyao Li

Associate Editor

PLOS Computational Biology

Jian Ma

Deputy Editor

PLOS Computational Biology

Reviewer's Responses to Questions

**Comments to the Authors:**

Reviewer #2: To address our criticism on the statistical formulation, the authors attempted to rephrase their goal as to find genes that share some correlated structure between two datasets ("interrelated genes"). However, this new goal is still not statistcally sound: the correlation structure the approximate distance correlation (ADC) aims to capture cannot be written down in the probability language using random variables.

To restate, this is what ADC does: given a tested gene, 1) selecting k top correlated genes in each dataset respectively; 2) pair the k top correlated genes by the order of Pearson correlation with the tested gene; 3) calculate Distance Correlation on the above k pairs.

We maintain our position that calculating the distance correlation between two sets of top correlated genes, which are selected respectively in each dataset and ordered by correlation, is biologically meaningless. This concern, if not addressed, would make the entire ADC method invalid.

1. A fundamental issue with the proposed ADC is that the input data of ADC are NOT the required input data of the distance correlation (DC). There are two unreconcilable differences:

(1) DC requires the input data to be naturally paired, and the pairing cannot be inferred from the same input data. In contrast, the paired "observations" input into ADC are inferred from the same input data. In other words, ADC uses the same input data for twice: first for inferring the pairing and second for calculating the DC. This would result in invalid, uninterpretable, inflated DC values.

(2) In ADC, the "observations" are NOT real observations but other genes. The authors' responses did not convince us that "correlated genes" of a given gene cannot regarded as observations of that one, an assumption that does not make either biological or statistical sense.

2. If the authors want to modify their paper to become statistically sound, we deem it necessary that the authors pair genes by external information (e.g., gene orthology), so that genes are naturally paired.

3. The new term "interrelated genes", which was modified from "similarly distributed genes", lacks both biological and statistical definitions.

Reviewer #4: In this paper, the authors propose Approximate Distance Correlation (ADC) to select interrelated genes with statistical significance across two different biological datasets.

Overall, I felt that the ADC method is meaningful and valid. The reviewer 2 made a good point that the previously-proposed concept "similarly distributed genes" is confusing and misleading. This has since been replaced by “interrelated genes” by the authors in this round of revision. In my opinion, the revisions made by the authors are responsive and adequate.

I think the ADC concept is valid and useful. But I felt that ADC is motivated more from the mathematical side, instead of the biological side. The results derived from ADC made sense, but I think other simpler measures may achieve similar results. Anyway, I think this is a very minor point. At this stage, the authors do not have to address it.

**Have the authors made all data and (if applicable) computational code underlying the findings in their manuscript fully available?**

Reviewer #2: None

Reviewer #4: Yes

PLOS authors have the option to publish the peer review history of their article (what does this mean?). If published, this will include your full peer review and any attached files.

Reviewer #2: No

Reviewer #4: No

---

## [Editor Report · Acceptance letter]

1 Nov 2021

PCOMPBIOL-D-21-00727R2 

Approximate distance correlation for selecting highly interrelated genes across datasets

Dear Dr Zhang,

I am pleased to inform you that your manuscript has been formally accepted for publication in PLOS Computational Biology. Your manuscript is now with our production department and you will be notified of the publication date in due course.

With kind regards,

Zsofia Freund
